# Evaluation of large language models as a diagnostic tool for medical learners and clinicians using advanced prompting techniques

Karolina Gaebe🆔, Benjamin van der Woerd🆔*

Division of Otolaryngology-Head and Neck Surgery, Department of Surgery - Michael G. DeGroote School of Medicine, Hamilton, Ontario, Canada

* vanderw@mcmaster.ca

## Abstract

### Background

Large language models (LLMs) have demonstrated capabilities in natural language processing and critical reasoning. Studies investigating their potential use as health-care diagnostic tools have largely relied on proprietary models like ChatGPT and have not explored the application of advanced prompt engineering techniques. This study aims to evaluate the diagnostic accuracy of three open-source LLMs and the role of prompt engineering using clinical scenarios.

### Methods

We analyzed the performance of three open-source LLMs—llama-3.1-70b-versatile, llama-3.1-8b-instant, and mixtral-8x7b-32768—using advanced prompt engineering when answering Medscape Clinical Challenge questions. Responses were recorded and evaluated for correctness, accuracy, precision, specificity, and sensitivity. A sensitivity analysis was conducted presenting the three LLMs with basic prompting challenge questions and excluding cases with visual assets. Results were compared with previously published performance data on GPT-3.5.

### Results

Llama-3.1-70b-versatile, llama-3.1-8b-instant, and mixtral-8x7b-32768 achieved correct responses in 79%, 65%, and 62% of cases, respectively, outperforming GPT-3.5 (74%). Diagnostic accuracy, precision, sensitivity, and specificity responses all outperformed those previously reported for GPT-3.5. Results generated using advanced prompting strategies were superior to those based on basic prompting. Sensitivity analysis revealed similar trends when cases with visual assets were excluded.

provided the original author and source are credited.

**Data availability statement:** All relevant data are within the manuscript and its Supporting information files.

**Funding:** The author(s) received no specific funding for this work.

**Competing interests:** The authors have declared that no competing interests exist.

## Discussion

Using advanced prompting techniques, LLMs can generate clinically accurate responses. The study highlights the limitations of proprietary models like ChatGPT, particularly in terms of accessibility and reproducibility due to version deprecation. Future research should employ prompt engineering techniques and prioritize the use of open-source models to ensure research replicability.

## Introduction

Large language models (LLM) are a type of artificial intelligence (AI) trained by deep neural networks on datasets that contain billions of parameters [1]. These models possess advanced capabilities in natural language processing, generation, and critical reasoning. One of the most well-known and easily accessible LLMs is Open AI's ChatGPT [2]. Other LLMs, such as Meta's LLaMA and Google's Bidirectional encoder representations from transformers (BERT) have been made available and shown potential to perform language processing tasks [3,4].

LLMs are increasingly gaining attention for their potential applications in health care as a diagnostic or educational tool [5,6]. Researchers demonstrated that ChatGPT can accurately answer biomedical and clinical reasoning questions on the United States Medical Licensing Examination (USMLE) at a level that approached or exceeded the passing threshold [7]. Several other studies have since investigated the ability of LLMs to respond to clinical scenario questions often with moderate accuracy [8–10]. These studies often cite limitations in the training dataset and lack of genuine understanding of the clinical scenarios as reasons for LLMs' suboptimal performances [9]. However, little attention has been paid to advanced prompting techniques. Furthermore, most investigations rely on a single LLM, typically GPT-3.5, without exploration of other models. One such study recently investigated the ability of ChatGPT to respond to clinical case scenarios. The authors found that GPT-3.5 had difficulty interpreting clinical findings and that responses lacked factual accuracy. They therefore concluded that ChatGPT has limited utility as a diagnostic tool [11].

The aim of the current study is therefore to explore the potential role for LLMs as an educational tool for medical trainees by using three LLMs and advanced prompt engineering techniques.

## Methods

This study aimed to replicate and expand upon prior research investigating the diagnostic accuracy and utility of large language models in clinical decision-making. We compared the performance of three LLMs (llama-3.1-70b-versatile, llama-3.1-8b-instant, and mixtral-8x7b-32768) against previously reported GPT-3.5 performance.

### Artificial intelligence models

Three open-source LLMs were selected for this study. Llama-3.1-70b-versatile and Llama-3.1-8b-instant are 70 billion parameter and 8 billion parameter models,

respectively, built by Meta that rely on a decoder-only transformer model architecture [12]. Mixtral-8x7b-32768 is a decoder-only model that uses a combination of eight 7 billion parameter models [13].

The models were chosen based on several key criteria. First, these LLMs are widely available and well-known open-source models, which aligns with our emphasis on reproducibility and accessibility. Second, they represent a range of model sizes, from small (8 billion parameters) to moderate (70 billion parameters), allowing us to investigate the impact of model size on performance. Third, the inclusion of models with slightly differing architectures allowed us to explore how architectural variations might influence diagnostic accuracy. Selecting models of varying sizes and complexities allowed us to assess the performance of smaller to medium-sized models with larger proprietary models in specific medical diagnostic tasks using high-quality prompting techniques.

Selecting models in this way serves our research objectives of accessibility and reproducibility. It also addresses broader concerns surrounding AI research and an overreliance on large, proprietary models. Through showcasing capabilities of more accessible models, we hope to promote future research and development in this direction, democratizing access to AI-assisted medical diagnostic tools.

### Input source

We tested the performance of three LLMs using previously described Medscape Clinical Challenge questions [11]. These questions are designed to challenge health care practitioners' diagnostic and treatment skills. Each question contains a clinical scenario with patient presentation, physical exam findings, laboratory test results, and relevant imaging findings. The user is prompted to provide a diagnosis based on the given information in a multiple-choice format [14]. After selecting a response, Medscape provides the user with feedback and information on the correct answer as well as the answer chosen by most users allowing for comparison of the model performance. According to Hadi et al., cases published between September 2021 and January 2023 without visual assets were selected [11].

The system prompt can be reviewed in full in the supplementary materials (Supplementary materials, page 2). It utilized several advanced prompting techniques previously reported in the literature and asks LLMs to select their answer from the Medscape provided multiple-choice options [15–18]. The prompt engineering techniques employed in this study were designed to optimize LLM performance in medical diagnosis tasks [18]. The prompt utilizes a multi-faceted approach, combining role-playing, task decomposition, and structured output formatting within an extensive markup language (XML) framework. It is based on advanced strategies such as Chain of Thought (CoT) analysis, knowledge integration, and self-consistency techniques to enhance the diagnostic reasoning [17,18]. Counterfactual thinking and meta-cognitive reflection are included to promote flexible diagnostic processes. The prompt emphasizes differential diagnosis and mimics real-world clinical decision-making by requesting additional information when necessary. This approach aims to elicit a thorough, well-reasoned, and transparent diagnostic process from the models, mirroring the complex cognitive processes involved in expert medical diagnosis. The structured XML format provides a clear hierarchy of information guiding the AI's analytical process and ensures machine-readability. It also facilitates easier evaluation and comparison of performance across different cases or models. By employing these advanced prompting strategies, we aimed to maximize the diagnostic capabilities of the LLMs and provide a standardized framework for assessing their performance in medical diagnosis tasks.

### Data collection

A simple javascript application was written to present each clinical case challenge to three separate LLMs simultaneously. In doing so, the base prompt was set the same for each language model. The responses were output to a CSV file for each model to a different column for review. The correct answers were extracted by KG with content reviewed by BV.

## Outcomes

The primary outcome was the percentage of cases for which each model provided the correct answer. Secondary outcomes were diagnostic accuracy and quality of medical information. The rate of true positives (TP), false positives (FP), true negatives (TN), and false negatives (FN) were recorded. Using these, accuracy ([TP+TN]/#total responses), precision (TP/[TP+FP]), sensitivity TP/[TP+FN], and specificity (TN/[TN+FP]) were calculated.

The quality of medical information was rated as either complete (the answer includes all relevant information for making an accurate diagnosis) or incomplete (the answer is missing some relevant information for making an accurate diagnosis) with the corresponding information described as being either relevant (the answer includes information that is directly relevant to the diagnosis) or irrelevant (the answer includes information that is not directly relevant to the diagnosis). Answers were then categorized as one of: complete/relevant, complete/irrelevant, incomplete/relevant, and incomplete/irrelevant. Ratings were undertaken by KG with answers verified by BV.

## Statistical analysis

Percentage of correct responses, overall diagnostic accuracy, precision, sensitivity, and specificity were summarized for each model in a narrative synthesis. Comparisons were made between the three LLMs and the previously published results using GPT-3.5, focusing on relative improvements in diagnostic performance. A sensitivity analysis using the same basic prompting techniques as previously reported to present cases to the LLMs was also performed (Supplementary materials, page 5). Given that GPT-3.5 is depreciated, an additional sensitivity analysis was conducted presenting advanced prompts to GPT-4.0 Turbo [2]. Chi-Square test for independence were performed to compare the number of correct responses between the three models selected by us and GPT-3.5 responses. An alpha of 0.05 was considered statistically significant. All statistical analyses were carried out in R (*base R*, version 4.2.1) [19].

During the data analysis, it was found that 128 studies contained images of which 43 provided the images without describing their content. Since Hadi et al. proposed excluding cases with visual assets, we performed a sensitivity analysis excluding clinical cases that did not provide a description of the image content [11].

## Results

### Correct response rate

Out of 150 cases, correct responses were provided for 119 (79%), 97 (65%), and 93 (62%) cases by llama-3.1-70b-versatile, llama-3.1-8b-instant, and mixtral-8x7b-32768, respectively. The answer provided by these LLMs aligned with the response given by the majority of Medscape users in 94 (63%), 90 (60%), and 96 (64%) cases, respectively. In all three instances, these three models outperformed responses provided by GPT-3.5 in the previous analysis (Table 1).

### Diagnostic accuracy

There was a total of 150 question with 4 multiple choice options per question. When considering rates of TP, FP, TN, and FN, llama-3.1-70b-versatile, llama-3.1-8b-instant, and mixtral-8x7b-32768 all outperformed GPT-3.5 ($p < 0.0001$, $p = 0.016$, $p = 0.048$, respectively; Table 1). Respective accuracy for these three LLMs was 90%, 81%, and 81% compared with 74% reported previously for GPT-3.5 (Table 1).

### Sensitivity analysis

Correct response rates were significantly worse when presenting llama-3.1-70b-versatile, llama-3.1-8b-instant, and mixtral-8x7b-32768 with question stems using basic prompting compared with advanced prompting techniques ($p = 0.0021$, $p < 0.0001$, $p = 0.00370$, respectively; Supplementary results, page 6; Table S1 in S1 Appendix). When comparing the responses generated from the three LLMs using basic prompting with previously reported GPT-3.5 results, only

**Table 1. Summary of LLM responses. P-values refer to Chi-Square tests performed between the LLMs selected by us and GPT-3.5 responses reported by Hadi et al. [11].**

| | llama-3.1-70b-versatile | llama-3.1-8b-instant | mixtral-8x7b-32768 | GPT-3.5[1] |
|---|---|---|---|---|
| Primary outcome assessment | | | | |
| Correct responses, n (%) | 119 (79) | 97(65) | 93 (62) | 74 (49) |
| P-value | <0.0001 | 0.0073 | 0.027 | Ref. |
| Response aligned with the response given by most Medscape users, n (%) | 94 (63) | 90 (60) | 97 (65) | 92 (61) |
| P-value | 0.81 | 0.81 | 0.63 | Ref. |
| Secondary outcome assessment | | | | |
| True positive, n (%) | 119 (20) | 97 (16) | 93 (16) | 73 (12) |
| False positive, n (%) | 31 (5) | 53 (9) | 57 (10) | 77 (13) |
| True negative, n (%) | 419 (70) | 390 (65) | 390 (65) | 373 (62) |
| False negative, n (%) | 31 (5) | 60 (10) | 60 (10) | 77 (13) |
| P-value | <0.0001 | 0.016 | 0.048 | Ref. |
| Accuracy, % | 90 | 81 | 81 | 74 |
| Precision, % | 79 | 65 | 62 | 49 |
| Sensitivity, % | 79 | 62 | 61 | 49 |
| Specificity, % | 93 | 88 | 87 | 83 |

LLM, large language models.

[1]GPT-3.5 responses as reported by Hadi et al. [11].

llama-3.1-70b-versatile outperformed GPT-3.5 (Supplementary results, page 6; Table S1 in S1 Appendix). When comparing the responses generated from the three LLMs with GPT-4.0 Turbo results using advanced prompting for all models, only llama-3.1-70b-versatile outperformed GPT-4.0 Turbo (Supplementary results, page 6; Table S2 in S1 Appendix).

Upon review of the cases, 22 did not contain any images in the case description or physical exam/investigation findings. 85 contained images and provided a description or analysis of the image contents. 43 cases contained images of either physical exam findings or investigations, but did not provide a description of the image contents. We performed a sensitivity analysis excluding clinical cases with images that did not provide a description of the imaging findings to mimic the inclusion criteria initially proposed. Correct responses were provided for 87/107 (81%), 70/107 (65%), and 72/107 (67%) questions by llama-3.1-70b-versatile, llama-3.1-8b-instant, and mixtral-8x7b-32768, respectively. GPT-3.5 showed a correct response for 64/107 (60%) questions (Table 2).

## Discussion

In this analysis of the performance of three different LLMs when answering clinical practice scenarios, all models achieved a high rate of accuracy using advanced prompting strategies. Response rates improved when cases with clinical images were excluded from the analysis. Our findings refute previous claims that LLMs are not accurate diagnostic tools and highlight the importance of prompt engineering. Future studies should focus on education around prompt engineering to optimize LLM usability.

In recent years, interest in the use of AI technologies in healthcare applications has risen [1]. Many studies have investigated the potential of LLMs in medical education and assessed their ability to accurately respond to clinical scenario questions. However, the majority of this research has relied on basic prompting strategies that did not fully leverage LLM capabilities [10,11,20–22]. Basic prompting involves presenting LLMs with simple, straightforward question that are often generic and provide minimal context. By contrast, advanced prompting employs prompt engineering to provide LLMs with more context, multi-step instructions, and often uses output format constrains or examples to optimize their performance on various

**Table 2. Summary of sensitivity analysis excluding cases that contained images but did not provide a description of the image content (n = 107). P-values refer to Chi-Square tests performed between the LLMs selected by us and GPT-3.5 responses reported by Hadi et al. [11].**

| | llama-3.1-70b-versatile | llama-3.1-8b-instant | mixtral-8x7b-32768 | GPT-3.5[1] |
|---|---|---|---|---|
| Primary outcome assessment | | | | |
| Correct responses, n (%) | 87 (81) | 70 (65) | 72 (67) | 64 (60) |
| P-value | 0.0006 | 0.40 | 0.26 | Ref. |
| Response aligned with the response given by most Medscape users, n (%) | 69 (65) | 69 (65) | 72 (67) | NR |
| Secondary outcome assessment | | | | |
| True positive, n (%) | 87 (20) | 70 (16) | 72 (17) | 64 (15) |
| False positive, n (%) | 20 (5) | 37 (9) | 35 (8) | 43 (10) |
| True negative, n (%) | 301 (70) | 279 (65) | 284 (66) | 278 (65) |
| False negative, n (%) | 20 (5) | 42 (10) | 37 (9) | 43 (10) |
| P-value | <0.0001 | 0.87 | 0.61 | Ref. |
| Accuracy, % | 91 | 82 | 83 | 80 |
| Precision, % | 81 | 65 | 67 | 60 |
| Sensitivity, % | 81 | 63 | 66 | 60 |
| Specificity, % | 93 | 88 | 89 | 86 |

LLM, large language models; NR, not reported.

[1]GPT-3.5 responses as reported by Hadi et al. [11].

tasks. These techniques allow LLMs to better understand the nuances of challenging tasks, such as medical case scenarios [15,18]. Response rates were significantly improved when using advanced prompting techniques, demonstrating that effective prompt engineering can optimize LLM performance and lead to significantly higher accuracy when answering clinical scenarios even when lower parameter models are used. Yuan et al. have shown that advanced prompting strategies can lead to a higher rate of correct responses and more succinct and relevant outputs when presenting GPT-4.0 Turbo with management questions in the field of gastrointestinal oncology [23]. Prompt engineering is a crucial skill for health care providers wanting to interact with AI systems, but frequently overlooked in the health care literature [15]. By better utilizing and understanding prompt engineering, we can realize the full potential of LLMs in transforming health care. Future research should therefore prioritize the development of educational resources and training programs focused on prompt engineering.

One example of prompt engineering is one- or multi-shot prompting. These approaches provide the model with a single or multiple examples of the desired task or output [24]. This understanding of the underlying tasks can better equip the model to carry out more complex tasks such as clinical reasoning. Most studies assessing the capabilities of LLMs in healthcare, however, have relied on zero-shot estimates where models are expected to perform tasks solely relying on their pre-existing knowledge and the prompt instructions without providing specific examples in advance [24]. Reports assessing ChatGPTs capabilities in healthcare applications frequently cite limitations in its training dataset as reasons for suboptimal results [9,25]. Specialized LLMs, such as BioBART which was pretrained on PubMed abstracts to deliver it with specialized knowledge in the biomedical domain or Med-PaLM tailored and tested on the medical domain, have been developed to overcome this limitation with demonstrated improvement in biomedical domain performance benchmarks [26,27]. However, as we have demonstrated in our study, one-shot prompting can lead to more accurate results even when smaller parameter models without specialized training datasets or GPT-4.0 Turbo are used. The assumption that LLMs will naturally perform well on medical tasks without tailored prompting overlooks the complexity of medical knowledge and reasoning. It also leads to an underestimation of their true capabilities. Future studies should focus on multiple-prompt techniques that provide the LLM with better contextual understanding to leverage their true capabilities.

ChatGPT is one of the first and most well-known LLM due to its public availability and easy accessibility via any browser [9]. Few studies investigating LLMs in healthcare have focused on other LLMs [9]. OpenAI regularly releases model updates and recently deprecated GPT-3.5 [28]. Since GPT is proprietary to OpenAI, deprecation of models implies that users will no longer be able to access these versions. This has significant implications when these models are used in research practice as future authors will no longer be able to replicate previous studies. In research, the ability to replicate findings is fundamental to validate study results and advance knowledge. Reliance on proprietary LLMs therefore limits long-term research advancement. On the other hand, open-source models like Llama and Mixtral that were employed in this research are more transparent on their underlying architecture and training data. These models also remain accessible over time allowing future researchers to replicate and verify study results. Future studies should therefore focus on open-source models that provide long-term accessibility.

This study has several limitations. First, since GPT-3.5 has been deprecated from the online platform and the application programming interface has reference to multiple versions of GPT-3.5 and the specific model is not referenced [11]. Therefore, we were unable to replicate previous study findings using our own algorithm. Second, all LLMs evaluated in this study were trained on publicly available datasets after the cutoff date for Medscape questions, which might give them marginal knowledge advantage over the previously assessed GPT-3.5. Third, our study was designed to demonstrate that LLM performance can be improved when using advanced prompting strategies. Therefore, we did not contrast different advanced prompting strategies. Future research should focus on assessing advantages and disadvantages associated with different prompt engineering techniques. Fourth, the Medscape Clinical Challenge questions, while designed for medical professionals, may not fully represent the complexity of real-world clinical scenarios. The questions are structured for online consumption and may inadvertently simplify or standardize clinical presentations. Furthermore, the use of multiple-choice questions as a proxy for clinical reasoning skills has inherent limitations. While such questions can assess knowledge recall and basic application, they may not fully capture the nuanced, iterative process of clinical decision-making that often involves gathering additional information, considering patient-specific factors, and weighing multiple potential diagnoses simultaneously.

Using AI in medical diagnosis and education raises important ethical considerations that must be carefully approached. Though our study demonstrates the potential of LLMs in accurately answering clinical questions, it is important to recognize that these tools should act as supplement or aid and, critically, not replace, human clinical judgment. Future research should explore the use of vision language models (VLMs) to understand and interpret clinically relevant images. This could significantly enhance the diagnostic capabilities of AI systems by allowing them to analyze visual data. Additionally, future projects could explore the potential of VLMs in medical education, examining how these might be used to train medical learners to interpret clinical images.

This study demonstrates LLMs can achieve high accuracy in responding to clinical practice scenarios when advanced prompting strategies are used. Future research should incorporate prompt engineering into their study design. Additionally, there is a need to address the limitations associated with overreliance on proprietary models like ChatGPT. Version deprecation, as seen with GPT-3.5, hinders accessibility and prevents the replication of research findings. To ensure long-term research viability and reproducibility, future studies should focus on open-source models that provide transparency and sustained accessibility over time.

## Supporting information

**S1 Appendix. LLM Writeup – Supp – 30 Mar 2025.**
(DOCX)

**S2 Appendix. LLM Questions – PLOS ONE.**
(XLSX)

## Author contributions

**Conceptualization:** Karolina Gaebe, Benjamin van der Woerd.

**Data curation:** Karolina Gaebe.

**Formal analysis:** Karolina Gaebe, Benjamin van der Woerd.

**Investigation:** Karolina Gaebe.

**Methodology:** Karolina Gaebe, Benjamin van der Woerd.

**Project administration:** Karolina Gaebe.

**Software:** Karolina Gaebe.

**Supervision:** Benjamin van der Woerd.

**Writing – original draft:** Karolina Gaebe.

**Writing – review & editing:** Karolina Gaebe, Benjamin van der Woerd.

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
