## [Decision Letter · Decision Letter 0]

PONE-D-24-46911Evaluation of large language models as a diagnostic tool for medical learners and clinicians using advanced prompting techniquesPLOS ONE

Dear Dr. van der Woerd,

Thank you for submitting your manuscript to PLOS ONE. After careful consideration, we feel that it has merit but does not fully meet PLOS ONE’s publication criteria as it currently stands. Therefore, we invite you to submit a revised version of the manuscript that addresses the points raised during the review process.

Thank you for your valuable submission. Your exploration of LLM using advanced and robust techniques is timely and thoughtful. Nevertheless, some concerns were raised requiring clarification and refinement.

Briefly, the authors need to clarify if testing followed the same scheme, and detail potential biases. Consider alternative testing approaches. For example, the authors present some assumptions about models and generalisation that are hard to validate without benchmarking. The debate of the limitations of advanced prompting detailing alternative approaches are important.

Another important aspect is the lack of addressing direct links with basic prompting. Please clarify and consider an additional condition or testing. For instance, report evidence to support that one-shot leads to better results and other examples of multiple-prompt techniques for robust findings and investigation.

The authors focused on three open-source, aligning with their objectives. However, it would strengthen to: 1. clarify selection criteria and factors like parameters and training; b. elaborate the strengths of each model and their efficiency and reliability.

Finally, consider cross-validating advanced and simpler prompting techniques to ensure the accuracy and reliability of your findings can be generalised.

Please respond to all comments and highlight the changes in the ms.

Wishing you success with the study. 

We look forward to receiving your revised manuscript.

Kind regards,

Thiago P. Fernandes, PhD

Academic Editor

PLOS ONE

Journal Requirements:

2. We note that your Data Availability Statement is currently as follows: “All relevant data are within the manuscript and its Supporting Information files.”

Please confirm at this time whether or not your submission contains all raw data required to replicate the results of your study. Authors must share the “minimal data set” for their submission. PLOS defines the minimal data set to consist of the data required to replicate all study findings reported in the article, as well as related metadata and methods (https://journals.plos.org/plosone/s/data-availability#loc-minimal-data-set-definition ).

If your submission does not contain these data, please either upload them as Supporting Information files or deposit them to a stable, public repository and provide us with the relevant URLs, DOIs, or accession numbers. For a list of recommended repositories, please see https://journals.plos.org/plosone/s/recommended-repositories .

Reviewers' comments:

Reviewer's Responses to Questions

Comments to the Author

1. Is the manuscript technically sound, and do the data support the conclusions?

Reviewer #1: Yes

Reviewer #2: Partly

2. Has the statistical analysis been performed appropriately and rigorously? 

Reviewer #1: Yes

Reviewer #2: Yes

3. Have the authors made all data underlying the findings in their manuscript fully available?

Reviewer #1: Yes

Reviewer #2: Yes

4. Is the manuscript presented in an intelligible fashion and written in standard English?

Reviewer #1: Yes

Reviewer #2: Yes

5. Review Comments to the Author

Reviewer #1: Clinically and pedagogically relevant and interesting study highlighting the limitations of proprietary models like ChatGPT,

particularly in terms of accessibility and reproducibility due to version deprecation.

Reviewer #2: The authors tested the performance of 3 open-source LLMs, 2 Llama 70b and 8b, and mistral 8x7b for answering Medscape Clinical challenge questions. They reported that the open-source LLMs performed better than GPT-3.5 with their prompting script. The investigation of prompting techinques is interesting but the evaluation/tests seem a little bin thin. I have the following major comments:

1. Comparison with GPT-3.5 is limited by the unavailability of GPT-3.5 for this testing. While the authors noted this in their limitation, this is a critical point they make about the superority of open-source LLMs with advanced prompting techinques. They should comment on whether they followed exactly the testing scheme as previously reported and what could bias their testing results if different. Alternatively, they could test the performance of GPT4.

2. The authors concluded that advanced prompting techinques could help LLMs generate accurate responses. However, there is no testing of basic vs advanced prompting. The testing of different prompting strategies is currently missing and therefore it is hard to draw the conclusion based on what is presented in the paper.

6. PLOS authors have the option to publish the peer review history of their article (what does this mean? ). If published, this will include your full peer review and any attached files.

Do you want your identity to be public for this peer review? For information about this choice, including consent withdrawal, please see our Privacy Policy .

Reviewer #1: No

Reviewer #2: No

---

## [Author Response · Author response to Decision Letter 1]

20 Dec 2024

Thank you for your valuable submission. Your exploration of LLM using advanced and robust techniques is timely and thoughtful. Nevertheless, some concerns were raised requiring clarification and refinement.

Briefly, the authors need to clarify if testing followed the same scheme, and detail potential biases. Consider alternative testing approaches. For example, the authors present some assumptions about models and generalization that are hard to validate without benchmarking. The debate of the limitations of advanced prompting detailing alternative approaches are important.

Another important aspect is the lack of addressing direct links with basic prompting. Please clarify and consider an additional condition or testing. For instance, report evidence to support that one-shot leads to better results and other examples of multiple-prompt techniques for robust findings and investigation.

The authors focused on three open-source, aligning with their objectives. However, it would strengthen to: 1. clarify selection criteria and factors like parameters and training; b. elaborate the strengths of each model and their efficiency and reliability.

Finally, consider cross-validating advanced and simpler prompting techniques to ensure the accuracy and reliability of your findings can be generalised.

Please respond to all comments and highlight the changes in the ms.

Wishing you success with the study.

Reviewer #1

Clinically and pedagogically relevant and interesting study highlighting the limitations of proprietary models like ChatGPT, particularly in terms of accessibility and reproducibility due to version deprecation.

Response: We thank the reviewer for their feedback.

Reviewer #2: The authors tested the performance of 3 open-source LLMs, 2 Llama 70b and 8b, and mistral 8x7b for answering Medscape Clinical challenge questions. They reported that the open-source LLMs performed better than GPT-3.5 with their prompting script. The investigation of prompting techinques is interesting but the evaluation/tests seem a little bin thin. I have the following major comments:

1. Comparison with GPT-3.5 is limited by the unavailability of GPT-3.5 for this testing. While the authors noted this in their limitation, this is a critical point they make about the superiority of open-source LLMs with advanced prompting techniques. They should comment on whether they followed exactly the testing scheme as previously reported and what could bias their testing results if different. Alternatively, they could test the performance of GPT4.

Response: We thank the reviewer for their comment. We confirm that that we followed the same testing scheme as previously reported with the exception that in our case a javascript was used to present clinical case challenges to the LLMs whereas the previous authors presented case challenges to GPT manually. Our approach eliminates subjective reviewer bias that may have occurred in the previous analysis due to individual prompting questions to ChatGPT.

We employed advanced prompting strategies to present case challenges to the LLMs. We agree with the reviewer that this makes it difficult to distinguish whether the change in prompting strategy or LLM lead to the difference in outcomes. We have therefore repeated our analysis using the basic prompting stem previously reported when presenting case challenges to the LLMs selected by us (Supplementary materials, page 6) to ensure benchmarking of testing approaches. We found superior results when using advanced prompting compared with basic prompting techniques when using the same LLMs. This supports our argument that prompt engineering is a crucial skills for clinicans wanting to use AI technologies.

2. The authors concluded that advanced prompting techinques could help LLMs generate accurate responses. However, there is no testing of basic vs advanced prompting. The testing of different prompting strategies is currently missing and therefore it is hard to draw the conclusion based on what is presented in the paper.

Response: We agree with the reviewer that this poses a limitation to our current analysis. We have therefore amended our analysis to include basic prompting with the same prompting stem as reported by the previous authors (Supplementary materials). As mentioned in our response to the previous comment, our findings confirm the superiority of advanced compared with basic prompting strategies. This confirms that there needs to be a greater emphasis on prompt engineering and generating education resources for prompt engineering in clinical research.

---

## [Decision Letter · Decision Letter 1]

PONE-D-24-46911R1Evaluation of large language models as a diagnostic tool for medical learners and clinicians using advanced prompting techniquesPLOS ONE

Dear Dr. van der Woerd,

Thank you for submitting your manuscript to PLOS ONE. After careful consideration, we feel that it has merit but does not fully meet PLOS ONE’s publication criteria as it currently stands. Therefore, we invite you to submit a revised version of the manuscript that addresses the points raised during the review process.

It appears that certain important adjustments were not fully addressed. The changes made seem minimal given the extent of the concerns raised. For example, the request to explore alternative methodological approaches was not sufficiently considered. Additionally, suggestions from the reviewers were not considered - which would have strengthened the study. Furthermore, clarification on the prompts and task design remains unclear. The theoretical context also requires further refinement, particularly re. the limitations, gaps, and strengths of the approach.

Overall, while the manuscript has potential, important changes are needed to address these issues.

Wishing you success with the study.

We look forward to receiving your revised manuscript.

Kind regards,

Thiago P. Fernandes, PhD

Academic Editor

PLOS ONE

Reviewers' comments:

Reviewer's Responses to Questions

**Comments to the Author**

1. If the authors have adequately addressed your comments raised in a previous round of review and you feel that this manuscript is now acceptable for publication, you may indicate that here to bypass the “Comments to the Author” section, enter your conflict of interest statement in the “Confidential to Editor” section, and submit your "Accept" recommendation.

Reviewer #1: All comments have been addressed

Reviewer #3: (No Response)

2. Is the manuscript technically sound, and do the data support the conclusions?

Reviewer #1: Yes

Reviewer #3: Partly

3. Has the statistical analysis been performed appropriately and rigorously? 

Reviewer #1: Yes

Reviewer #3: I Don't Know

4. Have the authors made all data underlying the findings in their manuscript fully available?

Reviewer #1: Yes

Reviewer #3: Yes

5. Is the manuscript presented in an intelligible fashion and written in standard English?

Reviewer #1: Yes

Reviewer #3: Yes

6. Review Comments to the Author

Reviewer #1: The study highlights the limitations of proprietary models like ChatGPT, particularly in terms of accessibility and reproducibility due to version deprecation.

Reviewer #3: The editor, and to a lesser extent the two reviewers, requested that you revise the paper. In my opinion, you have done this only rudimentarily. For example, you have not presented any alternative approaches as requested by the editor.

It is not clear to me why you did not follow Reviewer 2's recommendation to conduct the analysis with GPT-4. I would do this supplementary to GPT-3.5. This would make the comparison much more valuable.

Here are a few further remarks:

Although you included the prompts in the appendix, the text does not describe the difference between basic and advanced promptings. It is also not clear from the text that the LMS were required to choose a diagnosis from four options.

The theoretical part of the paper is very short. This may be appropriate for the topic and the current state of research. But, I recommend adding more about the state of the research question under investigation. For example, in the introduction, you do not explain why GPT does not provide good answers and why other LMS should. The title states that the use of LLMs for diagnosis is intended for the training of medical learners and clinicians. This is not clearly stated in the article, but it is important for ethical and practical reasons.

There is an error in the abstract (line 35): you report the 'correct response rate', but for GPT-3.5, the accuracy is reported (74% instead of 49%).

In line 93 of the methods section, you write "we demonstrate that even smaller to medium-sized models [...] can potentially outperform larger proprietary models". Shouldn't it say "we want to demonstrate"?

In line 163, you state that an "alpha of 0.5" is considered statistically significant. Shouldn't it say a p-value of 0.05 or 5%?

In line 164, you state that they analyzed with R (4.2.1). Today, it is standard to provide a reference and also mention and cite the packages used. This is completely missing in your paper."

"I think your article 'Evaluation of large language models as a diagnostic tool for medical learners and clinicians using advanced prompting techniques' is good and should be published soon. Unfortunately, I have a bit of a bad feeling about it. There are a few small but important points that are not correct (see above). These, along with the impression that more should be written in the introduction and methods sections, prevent me from recommending the article for publication as it is. It needs further revision.

I hope my comments will help."

7. PLOS authors have the option to publish the peer review history of their article (what does this mean? ). If published, this will include your full peer review and any attached files.

**Do you want your identity to be public for this peer review?** For information about this choice, including consent withdrawal, please see our Privacy Policy .

Reviewer #1: No

Reviewer #3: No

---

## [Author Response · Author response to Decision Letter 2]

27 Apr 2025

The editor, and to a lesser extent the two reviewers, requested that you revise the paper. In my opinion, you have done this only rudimentarily. For example, you have not presented any alternative approaches as requested by the editor.

It is not clear to me why you did not follow Reviewer 2's recommendation to conduct the analysis with GPT-4. I would do this supplementary to GPT-3.5. This would make the comparison much more valuable.

Response: We thank the reviewer for their feedback. In addition to the previously described sensitivity analysis using basic prompting for the three LLMs selected by us, we have provided results for a sensitivity analysis using advanced prompting strategies in GPT-4.0 Turbo (page 9 paragraph 1; supplementary materials, page 6, paragraph 3,4). Our findings demonstrate that advanced prompting strategies can lead to better results when applied to clinical practice scenarios even when smaller parameter models without specialized healthcare training or GPT-4.0 Turbo are used (page 13, paragraph 1).

Here are a few further remarks:

Although you included the prompts in the appendix, the text does not describe the difference between basic and advanced promptings. It is also not clear from the text that the LMS were required to choose a diagnosis from four options.

Response:

We thank the reviewer for their feedback and have provided additional context on the difference between advanced and basic prompting (page 12, paragraph 1). We have also clarified that LLMs were asked to select an answer from the Medscape multiple choice options (page 7, paragraph 1).

The theoretical part of the paper is very short. This may be appropriate for the topic and the current state of research. But, I recommend adding more about the state of the research question under investigation. For example, in the introduction, you do not explain why GPT does not provide good answers and why other LMS should.

Response:

We thank the reviewer for their suggestion. As stated in the introduction, the aim of our study was to demonstrate the role of LLMs as a diagnostic and educational tool using advanced prompt engineering. The focus was not to demonstrate inferior performance of ChatGPT, but rather show that using advanced prompt engineering, even smaller parameter LLMs can achieve accurate results. We chose open-source LLMs for this study for purposes of reproducibility. As discussed, our criticism on ChatGPT being used in previous studies is due to the fact that it is proprietary to OpenAI. Depreciation of previous ChatGPT models limits their access to future investigators and thus impacts reproducibility which is a cornerstone of research (page 13, paragraph 2). In fact, as shown in the sensitivity analyses, results using advanced prompting strategies were comparable between GPT-4.0 Turbo and the LLMs selected by us.

The title states that the use of LLMs for diagnosis is intended for the training of medical learners and clinicians. This is not clearly stated in the article, but it is important for ethical and practical reasons.

Response: As stated in the introduction, the aim of our study was to explore the role for LLMs as an educational tool (page 5, paragraph 1: “The aim of the current study is therefore to explore the potential role for LLMs as an educational tool for medical trainees by using three LLMs and advanced prompt engineering techniques.”).

There is an error in the abstract (line 35): you report the 'correct response rate', but for GPT-3.5, the accuracy is reported (74% instead of 49%).

Response: We have corrected this error in the abstract (page 2, paragraph 3).

In line 93 of the methods section, you write "we demonstrate that even smaller to medium-sized models [...] can potentially outperform larger proprietary models". Shouldn't it say "we want to demonstrate"?

Response: We have updated the text appropriately (page 6, paragraph 1).

In line 163, you state that an "alpha of 0.5" is considered statistically significant. Shouldn't it say a p-value of 0.05 or 5%?

Response: This error has been corrected (page 9, paragraph 1).

In line 164, you state that they analyzed with R (4.2.1). Today, it is standard to provide a reference and also mention and cite the packages used. This is completely missing in your paper."

Response: We have addended this information (page 9, paragraph 1).

---

## [Decision Letter · Decision Letter 2]

Evaluation of large language models as a diagnostic tool for medical learners and clinicians using advanced prompting techniques

PONE-D-24-46911R2

Dear Dr. van der Woerd,

We’re pleased to inform you that your manuscript has been judged scientifically suitable for publication and will be formally accepted for publication once it meets all outstanding technical requirements.

Kind regards,

Thiago P. Fernandes, PhD

Academic Editor

PLOS ONE

Additional Editor Comments (optional):

Reviewers' comments:

Reviewer's Responses to Questions

**Comments to the Author**

1. If the authors have adequately addressed your comments raised in a previous round of review and you feel that this manuscript is now acceptable for publication, you may indicate that here to bypass the “Comments to the Author” section, enter your conflict of interest statement in the “Confidential to Editor” section, and submit your "Accept" recommendation.

Reviewer #3: All comments have been addressed

2. Is the manuscript technically sound, and do the data support the conclusions?

Reviewer #3: Yes

3. Has the statistical analysis been performed appropriately and rigorously? 

Reviewer #3: Yes

4. Have the authors made all data underlying the findings in their manuscript fully available?

Reviewer #3: Yes

5. Is the manuscript presented in an intelligible fashion and written in standard English?

Reviewer #3: Yes

6. Review Comments to the Author

Reviewer #3: Thank you for addressing all requested propositions.

I wish you that the paper will be read and cited often.

7. PLOS authors have the option to publish the peer review history of their article (what does this mean? ). If published, this will include your full peer review and any attached files.

**Do you want your identity to be public for this peer review?** For information about this choice, including consent withdrawal, please see our Privacy Policy .

Reviewer #3: No

---

## [Editor Report · Acceptance letter]

PONE-D-24-46911R2

PLOS ONE

Dear Dr. van der Woerd,

I'm pleased to inform you that your manuscript has been deemed suitable for publication in PLOS ONE. Congratulations! Your manuscript is now being handed over to our production team.

Kind regards,

on behalf of

Dr. Thiago P. Fernandes

Academic Editor

PLOS ONE